# Development of a conceptual framework of sexual health for married elderly women: An exploratory mixed method study

**Farzaneh Saadatmand[1], Mitra Savabi-Esfahani[2], Zahra Heidari[3], Marjan Beigi** [iD][2]*

**1** Student Research Committee, Faculty of Nursing and Midwifery, Isfahan University of Medical Sciences, Isfahan, Iran, **2** Department of Midwifery and Reproductive Health, Faculty of Nursing and Midwifery, Isfahan University of Medical Sciences, Isfahan, Iran, **3** Department of Biostatistics and Epidemiology, Faculty of Health, Isfahan University of Medical Sciences, Isfahan, Iran

* beigi@nm.mui.ac.ir

## Abstract

### Background

Identifying elderly women's sexual health demands based on their lived experience is essential since it leads to their sexual rights promotion. The present framework was designed to identify sexual demands, develop a reliable and valid questionnaire, and present it to policymakers.

### Methods

This conceptual framework was designed in four phases using an exploratory sequential approach. The first phase included a review of related literature to identify elderly women's sexual health demands. In the second phase, the qualitative phase, the participants will be interviewed to identify their sexual health demands. Data will be collected through in-depth semi-structured interviews until data saturation. In the third phase, a draft questionnaire will be provided using the data from the qualitative study and literature review to measure the sexual health demands of elderly women. Afterward, the face and content validity, as well as the construct and criterion validity, will be determined. The construct and criterion validity will be assessed using exploratory factor analysis and a cross-sectional study on elderly women, respectively. Finally, the reliability of the questionnaire will be determined. In the fourth phase, using the classical Delphi technique, the panel of experts will select the most important demands to present to policymakers.

### Results

Based on a literature review conducted from October 20, 2023, to March 15, 2024, the demands were categorized into three main groups: "the elimination of ageist attitudes," "the elimination of sexism," and "the elimination of sexual retirement." Other data and their integration with the literature review data to finalize the categories will be obtained after conducting qualitative interviews. Afterward, the questionnaire will be developed and psychometrically evaluated. Finally, the demands will be prioritized for policymakers.

**Data availability statement:** This manuscript is a study protocol and we currently do not have any data. The results presented in the manuscript (Protocol) pertain to the literature review conducted and are not based on any data generated within the scope of this study. All relevant data will be accessible after the study has been completed and published.

**Funding:** The author(s) received no specific funding for this work.

**Competing interests:** The authors have declared that no competing interests exist.

## Conclusion

According to the results, elderly women face social and cultural challenges in maintaining sexual health. Based on the presented methods, comprehensive and complete identification of demands in different cultures can facilitate sexual health planning.

## Introduction

Increased life expectancy worldwide and improved access to healthcare services have caused a noticeable demographic change. Consequently, most countries will face population aging in the coming years. Older people experience the most rapid population growth rate. In 2012, this population was 13.7%; it is expected to be approximately 20% in 2050. This increase is beyond the growth of the entire population [1]. Moreover, women live longer on average than men. It suggests that more elderly women will require help to deal with age-related changes and maintain their quality of life [2]. Currently, approximately two-thirds of the world's older population lives in developing countries. In Iran [3], it is expected to be 14%. It is estimated that the Iranian elderly population will reach approximately 30% in 2050 [4]. However, despite the increase in life expectancy, healthy life expectancy has not equally increased. This implies that many reach this stage with health problems impairing their quality of life [5]. As one of the most imperative societal issues, individuals' health should be given special attention. It is because an unhealthy society lacks the necessary dynamism to progress and advance its goals. Moreover, overlooking this issue will place a heavy burden on society. Sexual health is part of general health. Sexual identity, sexuality, sexual orientation, behaviors, and functions, and the subjective and multifaceted concepts of human sexuality are directly related to public health [6]. According to the definition of the World Health Organization, sexual health means physical, mental, and social health concerning sexuality. It does not merely mean the absence of disease, disorder, or weakness [7]. Sexual health requires a positive and deferent approach to gender, sexual relations, and individuals. It also needs pleasurable and safe sexual experiences without coercion, discrimination, and violence [8]. Sexual health enhances the individual's skills in emotional dialogues and decision-making regarding the type of sexual behavior. It also empowers them to improve and maintain sexual function [9]. Sexual health is considered an essential indicator of older people's quality of life. However, despite this prominence, most sexual health needs remain unmet [10]. According to evidence, many elderly women face difficulties in receiving sexual health services since healthcare workers are more focused on screening to identify their physical illnesses [11]. Moreover, women's sexual function after reproductive years is assumed unimportant under the influence of sexual schemas [12]. Paying closer attention to improving elderly women's sexual health, as one of the subsets of public health, seems necessary. That is because a sexually healthy society has a higher public health index, and healthy individuals will participate more dynamically in their personal and social lives [13]. Measuring and determining personal demands requires an efficient tool based on society's culture and social norms. Reviewing the related literature shows that designed tools address different areas of women's sexual health, including sexual functioning [14], sexual satisfaction [15], sexual behaviors [16], sexual disorders [17], sexual health literacy [18], and other areas in youth and adulthood. A limited number of studies have evaluated sexual health demands in older people. Therefore, due to the scarcity of studies, inadequate knowledge about elderly women's sexual health [19], and women's negative sexual socialization [20], identifying sexual health demands is necessary and will facilitate health policymaking. Such research is needed for health policymakers in Iran and other countries. That is because these countries lack intermediary institutions or supporting non-governmental organizations for health services. In these countries, these policymakers establish health-related laws. In addition, health programs are also implemented and defined by these policymakers and

legislators. This conceptual framework aims to assess elderly women's sexual health demands. After identifying these demands, a specific, valid, reliable, and culturally and socially appropriate tool will be designed and developed, and demands will be prioritized for policy making.

## Materials and methods

### Ethics approval

This study is supported by the Isfahan University of Medical Sciences. The Ethics Committee of the Isfahan University of Medical Sciences in Isfahan, Iran, approved the protocol of this study (code number: IR.MUI.NUREMA.REC.1401.157).

### Study design and setting

This is an exploratory mixed-method, multi-phase study to identify the concept of elderly women's sexual health demands. Both qualitative and quantitative methods will be used; therefore, the paradigm in this research is pragmatic [21]. In this study, a review of the literature was carried out, and a qualitative study will be conducted. Afterward, based on the data obtained in the qualitative phase and the literature review, a questionnaire for measuring elderly women's sexual health demands will be prepared. After designing the questionnaire, its face, content, criteria, and construct validity will be determined. The reliability of the questionnaire will also be determined. Eventually, using a decision matrix, a panel of experts will select the most important demands to present to policymakers (Fig 1).

### The specific objectives of each phase

- Phase one aims to identify elderly women's sexual health demands based on a literature review.
- Phase two aims to explain elderly women's and service providers' experiences regarding elderly women's sexual health demands.

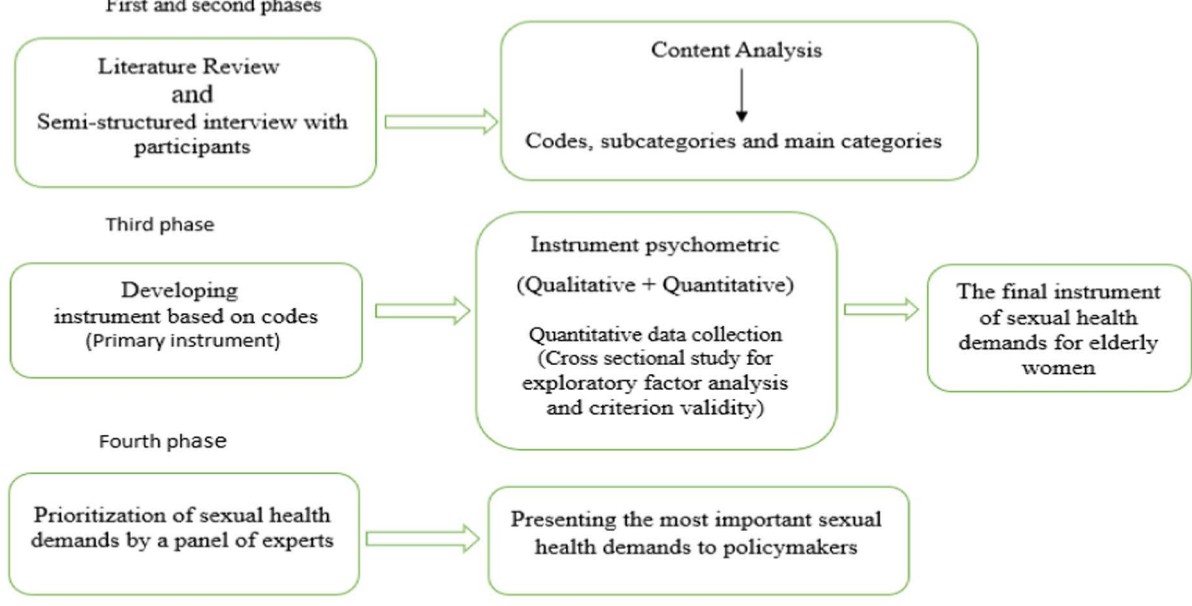

**Fig 1. Study design diagram.**

- Phase three aims to develop appropriate items for the initial version of the sexual health demands assessment tool for elderly women, determine the qualitative and quantitative face validity of the sexual health demands assessment tool for elderly women, determine the qualitative and quantitative content validity of the sexual health demands assessment tool for elderly women, determine the construct validity of the sexual health demands assessment tool for elderly women, and determine the criterion validity of the sexual health demands assessment tool for elderly women.

- Phase four aims to prioritize demands and present them to policymakers

## 1. First phase: Reviewing the literature

This stage aims to review studies to understand and explain the consistencies or inconsistencies of other studies with the present study. It also aims to enrich the findings of the qualitative section of the study in order to obtain comprehensive information. At this stage, the researcher searched for information related to elderly women's sexual health demands using reliable national and international databases. To review the literature, the Matrix Method was used based on Judith Garrard's guidebook. This method consists of creating four folders: an activity registration folder, a document folder, a review matrix folder, and a synthesis folder. In this study, these four folders were created; scientific literature was reviewed [22]. The steps of the matrix method are as follows:

- Creating an activity registration folder: All keywords used in relation to internet searches were listed in this folder. To review the literature related to elderly women's sexual health demands, the keywords sexual health, old age, old woman, Conceptual framework, aging, sexuality, demands, demands assessment tool, and their Persian equivalents between 2000 and 2022 were searched in Google Scholar, Scopus, Pubmed, Web of Science, Cochrane, ProQuest, CINAHL, Iran Medex, Magiran, and SID databases.

- Document folder: This folder contains all documents that were selected, downloaded, and saved during the search. In this study, the article abstracts were read after selecting keywords and conducting database searches. If they were aligned with the research objectives, they were selected; otherwise, they were discarded. Inclusion criteria included quantitative and qualitative articles in English or Persian, while exclusion criteria included articles written as brief notes or letters to the editor. The STROBE and COREQ checklists were used to assess quantitative [22] and qualitative studies [23], respectively. Articles that meet the quality assessment threshold on the STROBE checklist (15.5) and the individuals meeting all COREQ checklist criteria were included in this study.

- Matrix folder: In this stage, three researchers extracted data from articles that were evaluated and accepted using the STROBE and COREQ checklists. The extracted data included the study characteristics, such as author, year of publication, study design, sample size, study method, participants' characteristics, and research results. After extracting the data, they were reviewed and summarized in tables or text pages. It was performed by the same three researchers based on mutual agreement.

- Synthesis folder: Critical analysis of the texts was conducted during the synthesis stage to clarify the objective of the matrix method. Since the methodologies of the reviewed articles are not similar and both quantitative and qualitative articles were included in the research, a thematic summary approach was used to integrate the data.

It should be noted that, the review of the studies began from the beginning of writing the research plan; however, after the completion of the qualitative research and the collection of

study data, the review of the studies will be more structured and will continue until the sexual health demands of elderly women are presented.

## 2. Second phase: A qualitative study

This phase aims to explore the experiences of elderly women and service providers concerning the demands related to their sexual health.

### Study participants and sampling

The participants in this phase of the study will include two groups: women and healthcare service providers willing to participate. Healthcare service providers will include all specialists (reproductive health specialists, counselors, and psychologists) working in the field of sexual counseling. The first group of participants (elderly women) will be selected using purposeful sampling and will continue with a sampling strategy with maximum diversity in terms of age and social and economic status, marriage duration, and marital stability. The second group of participants (healthcare services providers) will also be selected purposefully. Their sampling will continue using maximum diversity in terms of work experience and career field (all specialists with different fields of study and professional experience in the field of sexual counseling). Informed consent will be obtained from all participants. The researcher will provide participants with explanations about the research objectives and data collection and protection. The participants will also be given the option to withdraw and leave the study at any stage. The participants' rights, reverence, and dignity will be observed in this study. Researchers will adhere to the Declaration of Helsinki for ethical considerations, and written consent will be obtained. Key items in this consent form include the following: Participants may withdraw from the study at any time, either at the beginning or during the interview, for no reason. Recorded interviews will be kept confidential and deleted from the recording device after the researcher analyzes them. Participants will be assured that their care at health centers will not be affected if they do not participate in the study. In order to select the samples, the researcher will attend the comprehensive healthcare centers, sexual health clinics, counseling centers, and personal offices of reproductive and sexual health specialists and midwives. She will then invite the study samples that meet the inclusion criteria for an interview.

Before that, the researcher will obtain an introduction letter from the university's research department (where the ethics code was approved) to provide to health center authorities and specialists. Permission to interview elderly women will be obtained from health center authorities. Specialists may also participate in interviews separately. The inclusion criteria for elderly women will include being married, not suffering from mental or physical illnesses affecting sexual function, and being able to participate in the study mentally and physically. The exclusion criterion will be the willingness to withdraw from participation at the beginning or during the interview. The inclusion criterion for specialists will be being active in the field of sexual health for two or more years. The exclusion criterion for them will be the willingness to withdraw from participation at the beginning or during the interview.

### Data collection tool and method

Data will be collected through in-depth and semi-structured interviews. In addition to interviews, field notes will also be used. The interviews will be conducted in the participants' preferred places and recorded using an mp4 device. Data collection will continue until data saturation. The interview questions for elderly women will be: How would you describe your sexual experiences at this stage of life (old age)? And what needs and demands do you have to maintain sexual health during this period? The questions for health service providers will

be: What do you think are the sexual health needs and demands of elderly women? And, as a health service provider for the elderly, what role do you have in meeting their sexual health demands? Interviews will begin with these questions, and then participants' open and interpretive responses will guide the process. To clarify participants' explanations, the researcher will use probing questions (e.g., What do you mean? Please give an example or explain more).

## Data analysis

Conventional qualitative content analysis will be used to analyze the data. This process is based on the conventional content analysis model by Graneheim and Lundman. This approach includes the following steps: determining the content of the analysis, defining the unit of analysis and initial coding, categorizing codes into subcategories, and forming the main category from subcategories [24]. All steps of this analysis, from the transcription of interviews word for word to the compilation of codes, subcategories, and main categories, will be carried out by two researchers. The researchers will reach a consensus to finalize each step. In case of any disagreement at any stage of the process, the opinion of the more experienced researcher will prevail.

## Rigor and trustworthiness

In order to investigate the correctness of the data, the criteria proposed by Lincoln and Guba, namely credibility, dependability, transferability, and confirmability, will be used [25]. In order to obtain credibility, various methods will be used: allocating enough time and prolonged engagement for data collection, triangulation, including in-depth interviews, note-taking, and selecting participants with maximum diversity. To obtain dependability, initial codes and examples of extracting categories, themes, and items from the interview texts for each category will be provided to an external observer. With regard to confirmability and in order to confirm and interpret the codes, the codes and themes will be shared with several experts in two sessions, and their comments will be applied as a member check. In order to increase transferability, the study findings will be provided to individuals with similar characteristics outside the study to judge whether the study results are similar to their own experiences.

## 3. The third phase: Designing and evaluating psychometrics of the tool for measuring elderly women's sexual health

At this phase, based on the findings of the qualitative phase and the information obtained from the literature review, the questionnaire's items and expressions will be designed. Moreover, its psychometric characteristics will be determined. The tool for assessing the sexual health demands of elderly women will be designed using the method proposed by Creswell and Clark (2011). In this method, the following steps will be used: Determining what should be measured, producing an item bank, determining the scale for the items and the physical structure of the instrument, considering valid items in other instruments to include in the tool, reviewing the item bank by experts, implementing the questionnaire on one sample for validation, evaluating items in terms of correlation between item-tool, reliability, optimizing the length of the tool based on the item performance, reviewing several samples of exploratory mixed-method studies on instrumentation, and the final formation of the tool [26]. Before the sexual health questionnaire, the common variables gathered from the literature review, such as age, age difference with the spouse, age at marriage, educational level (illiterate, primary, secondary, higher), occupation (housewife, employed), and duration of marriage, are considered.

## Instrument validity

In this study, the instrument's validity will be investigated using face, content, construct, and criterion validity.

## Face validity

Qualitative and quantitative methods will be used to determine face validity. To measure the qualitative face validity, Ten elderly women will be individually interviewed about the simplicity, fluency, and comprehensibility of the items, and their opinions will be applied to the questionnaire. In order to quantitatively assess the face validity, the item impact score will be calculated [27]. To this end, firstly, the opinions of 20 experts (reproductive health, psychologists, and counselors) about the importance of each statement will be sought on a 5-point Likert scale as very important (5), important (4), relatively important (3), slightly important (2), and unimportant (1). The item impact score will be obtained by multiplying the percentage of participants evaluating the phrase as important. In other words, they will have scored the item as 4 or 5 (frequency). The importance score of each phrase will be obtained by the mean. If the item's impact score is equal to or greater than 1.5, the item will be retained in the tool.

## Content validity

The tool's content validity will be investigated in the quantitative and qualitative stages. During qualitative content validity, the tool items' comprehensibility will be investigated by 20 experts (reproductive health, midwife, psychologist and consultant) in terms of grammar and appropriate wording and expressions. The experts will be requested to comprehensively express their written, corrective opinions [26].

Quantitative content validity will be investigated in two stages: Content Validity Ratio (CVR) and Content Validity Index (CVI).

**Content validity ratio (CVR).** For the content validity ratio, 20 experts will be requested to determine the necessity of the intended phrase for operationalizing the concept under study (3 = necessary, 2 = useful but not necessary, and 4 = not necessary). In this study, CVR will be calculated based on Lawshe's formula. The numerical value of the content validity ratio will be calculated using the following formula: $CVR = (Ne-N/2)/(N/2)$ [28].

In this formula, NE is the number of experts who evaluate the issue as necessary; N is the number of experts. If the CVR for each term is less than the numerical value in the Table, that item will be removed. Considering the number of expert team members in the present study will be 20, the minimum CVR value will be considered 0.42 [29].

**Content validity index (CVI).** In order to calculate this index, 20 experts will be requested to evaluate each item's content relevance, clarity, and simplicity.

The content validity index will be calculated using the formula of $CVI = n/N$.

CVI = content validity index, n = the number of experts scoring an item as 3 and 4, and N = the total number of experts who answer that item [30].

## Initial test execution

In this phase, the designed questionnaire, whose face and content validity have been assessed, will be initially administered to a limited number of individuals from the target group. The purpose of this initial test execution is to ensure the questionnaire has an appropriate structure; therefore, questions with a low Cronbach's alpha coefficient (below 0.75) will be removed at this stage. The sample size for conducting the pilot study will be 10% of the total sample size required for construct validity testing [31].

## Construct validity

Construct validity specifies the adequacy of the instrument to examine existing constructs. Exploratory factor analysis is a method for determining construct validity. This analysis is used to summarize and categorize data into correlated groups when little information is available about the construct under study [32]. In this study, exploratory factor analysis will be used to determine the relationships between concepts and presumed relationships between the items and their underlying dimensions. In this regard, a descriptive cross-sectional study will be conducted to psychometrically measure the tool.

Sampling, sample size, and data analysis.

The study population at this stage will include elderly women referring to comprehensive health centers. Sampling will be performed using the convenient method in these centers. To minimize bias in this non-random sampling approach, comprehensive health centers for sampling elderly women will be selected randomly. For this purpose, the city of Isfahan will be divided into four regions based on the socioeconomic status. One comprehensive health center will be randomly chosen from each region (using a lottery), and research participants from the selected centers will be then sampled using a convenience sampling method.

Inclusion criteria will include willingness to participate in the study, informed consent to provide information, women of Iranian nationality with a spouse, the ability to understand questions or having the least literacy level, and no mental and physical disorders affecting sexual function (based on the participants). The exclusion criteria will include unwillingness to continue cooperation in the study and scratched answers in the questionnaire.

In order to determine the sample size in the studies, the minimum and the maximum ratio of the sample to the item is 3 and 10 participants, respectively. In the present study, the sample size will be determined based on the items of the tool designed following the initial qualitative study. The sample-to-item ratio will be at least 5:1.

To assess the adequacy of the sample size and the correlation between extracted factors, the KMO (Kaiser-Meyer-Olkin) test and Bartlett's test of sphericity will be used. Then, exploratory factor analysis with varimax rotation will be employed to identify interpretable factors. A KMO value greater than 0.7 is interpreted as acceptable. Besides, a large sample size is suitable for exploratory factor analysis. Bartlett's test of sphericity should yield significant results ($p < 0.05$). To determine the best structure, an eigenvalue greater than one and a factor loading equal to or greater than 0.4 will be applied [31].

## Criterion validity

In this research, concurrent validity, which is one of the types of criterion validity and indicates whether a test can be a suitable substitute for another test (without delay and simultaneously), will be used.

Sampling, sample size, and data analysis.

A total of 140 elderly women referring to comprehensive health centers will be considered for this stage. Comprehensive health centers will be considered randomly, and the samples will be selected non-randomly using the convenient sampling method. To perform concurrent criterion validity, the Sexual Health Questionnaire for Middle-Aged Women by Moghasemi et al. (2022) will be utilized. This questionnaire includes 39 items, and it is rated on a Likert scale (1: strongly disagree - 5: strongly agree) in six subscales (sexual health care, holistic multidimensional sexual preparation, quality of sexual relationship, partner sexual incompetency, compatibility and resolving sexual problem conflict, and conservative socio-cultural norms) [33]. Although Moghasemi et al.'s questionnaire and the questionnaire of the present study differ in the target population, they are similar in subject matter, making it logical to compare

them. It should be noted that Moghasemi et al.'s questionnaire is the only one available to compare with the questionnaire of the present study.

In this regard, the correlation coefficient between the scores obtained from the designed questionnaire will be calculated using the Moghassemi tool. The greater these coefficients are, the more valid the study will be [29].

## Instrument reliability

The mean content validity index of each item equal to or above 0.78 will be accepted, and the items receiving a lower value will be removed. To investigate the reliability of the tool, internal consistency and stability will be determined. Cronbach's alpha coefficient will be used to determine the internal consistency of different questionnaire sections. To this end, the questionnaire will be distributed among 20 eligible elderly women and will be analyzed using SPSS software. Moreover, its Cronbach's alpha coefficient will be determined in different domains. The acceptable consistency coefficient will be equal to or greater than 0.75.

In order to determine the stability, the test-retest method will be used. For this purpose, the designed questionnaire will be provided to 20 eligible elderly women twice, at an interval of two weeks. Afterward, reliability and stability will be determined by calculating and comparing the Pearson correlation coefficient [31].

## 4. The fourth phase of the study: Presenting sexual health demands to policymakers

In this phase, the most important demands will be evaluated and prioritized to be presented to the policymakers at the Ministry of Health and Medical Education. This process will be carried out using a panel of experts and the classic Delphi technique in two in-person rounds. Fifteen presenters and experts in geriatric health from Isfahan University of Medical Sciences will be considered as panel members. In the first round, a questionnaire consisting of two parts will be given to the experts. The first part assesses sexual health demands based on the three criteria of " acceptability," "proportionality," and "runnability," while the second part includes the experts' written comments and suggestions. The demands will be evaluated using these two parts. In the second Delphi round, the panel members will be invited back to prioritize the evaluated demands. The prioritization will be done based on the selection matrix tables, and the demands will be presented to policymakers at the Ministry of Health and Medical Education. Since presenting the demands to the policymakers is for developing health plans, these policymakers will be encouraged to collaborate with the researchers to strengthen these plans. It will also be communicated to policymakers that researchers can actively participate in their meetings as key informants, assisting in the implementation of sexual health demands for elderly women.

## Results

Out of the 83 studies (quantitative and qualitative) found in the field of sexual health of the elderly, nine studies were selected for final review [34–42]. These studies were reviewed and examined between October 20, 2023, and March 15, 2024. The most important demands based on the studies were categorized into three main groups: the elimination of sexism, the elimination of ageist attitudes, and the elimination of sexual retirement. The subcategories for the elimination of sexism include the elimination of sexual superiority for men and the emphasis on equal satisfaction in sexual activities.

According to studies, sexism is linked to cultural deficiencies in societies. This type of discrimination is a significant barrier to women attaining sexual rights. Achieving sexual

rights based on respect for human rights is a factor in eliminating male sexual dominance and emphasizes equal sexual consent between partners. Educational and promotional interventions on this topic, from the healthcare system as well as non-governmental organizations (such as women's rights associations), are considered strategies for addressing sexism.

The subcategories of elimination of ageist attitudes include eliminating youth-oriented traditions, focusing on the high potential of elderly individuals with sexual health, and eliminating sexual taboos for the elderly.

According to studies, ageism in societies arises from inequality and discrimination between younger and older age groups. Elderly individuals need access to services to improve their sexual health; however, youth-centered traditions and the taboo around sexual activity for seniors hinder it. This problem stems from misguided attitudes among the public and healthcare system employees. Focusing on improving intergenerational connections through social media and strengthening senior health service programs are regarded as effective strategies to eliminate ageist perspectives.

The subcategories for eliminating sexual retirement include enabling the enjoyment of sexual pleasure in old age, strengthening marital bonds through the continuity of sexuality, and improving sexual interactions through the persistence of sexuality.

Reviewed studies indicate that older women have the motivation to maintain and engage in sexual activities. They can enjoy sexual activities and feel empowered and at peace afterward. Additionally, by engaging in sexual activities, they experience greater sexual desire, which subsequently enhances their emotional and marital interactions. Strategies for eliminating ageist attitudes and sexism contribute to terminating "sexual retirement" and enabling older women to achieve sexual health. Access to additional data and their integration will occur after qualitative interviews. Following that, the questionnaire will be developed and psychometrically evaluated. Finally, the extracted demands will be prioritized and sent to policymakers for policy formulation.

## Discussion

The present framework was designed to identify the concept of elderly women's sexual health demands. Consequently, a reliable and valid questionnaire will be developed, and the demands will be presented to policymakers. According to the results of this study, elderly women's sexual health demands are rooted in social and cultural issues, and the need to eliminate sexism highlights the existence of discrimination and sexual stereotypes in societies. The need to eliminate ageist attitudes indicates social discrimination and cultural deficiencies. Age and sexual discrimination in the elderly reduce productivity and abilities in marital life, forcing the elderly into sexual retirement. Hashemiparast et al. (2021) from Iran, in alignment with reviews on this conceptual protocol, identified age discrimination as the most significant barrier to elderly women's sexual health [43]. Okiria (2014) from Africa emphasized age and sex discrimination as key factors reducing sexual activity, thus affecting elderly women's sexual health. This researcher also noted that these women still desire to express sexual feelings and continue sexual activities despite the discrimination arising from societal attitudes internalized by elderly couples [44]. Erens et al. (2019) indicated that in British culture, physical and hormonal limitations due to aging decrease sexual desire and activity in the elderly [45]. Stentagg et al. (2021) from Sweden reported that while elderly men are more sexually active than elderly women, women experience greater satisfaction with their sexual health. Moreover, a reduction in their sexual activity is linked to decreased biological issues following sexual hormone reduction [46]. There are both similarities and differences in these studies. A common finding across African, Iranian, Swedish, and British studies is the decline of sexual life among women. However, these studies vary in the factors causing decreased sexual activity

and resulting sexual health issues. British and Swedish cultures view physical problems due to hormone decline and aging as contributing factors to sexual health issues. However, African culture emphasizes age and sex discrimination, and Iranian culture views age discrimination as a factor affecting sexual health. The primary cause of discrimination in Iran and Africa hindering elderly women's sexual health goals is the persistence of women's subordinate sexual roles—a deep-rooted issue that policymakers have not addressed. The results, however, show that elderly individuals are capable of maintaining sexual health, which in turn strengthens emotional and marital bonds and fortifies family relationships. Based on the review findings, emphasis should be placed on ensuring and enhancing the sexual health of the elderly. This is because sexual health is part of overall health and essential for achieving millennium development goals. This represents a harmonious, intimate, and affectionate married life where individuals enjoy a healthy, appropriate, and normal relationship.

Ensuring the elderly's sexual health, particularly in Iran and culturally similar regions such as South Asia and Africa, is vital since the elderly, especially women, in these countries are not sent to nursing homes and remain closely connected with their children and grandchildren. Therefore, disruptions in elderly health can lead to disruptions in youth health and the community [47,48]. Accordingly, preventive, supportive, and therapeutic strategies are suggested by media outlets, local cultural centers, and health centers. Health centers can assist by offering uniform sexual health services to both youth and the elderly. Local cultural centers and media can play a key role in addressing the elimination of age and sex discrimination. These measures could foster a culture of eliminating "sexual retirement" among elderly women. To implement these strategies, the Ministries of Health and Welfare should coordinate to provide integrated actions to ensure elderly women's access to sexual health.

This protocol (developing a valid and reliable questionnaire and presenting demands of sexual health to policymakers) can improve older people's quality of life. Using the questionnaire of the present study, the sexual health demands of elderly women can be identified in clinics and comprehensive health centers, allowing for the development of strategies or immediate solutions for these women. Presenting demands of sexual health to policymakers and legislators can also help them carry out efficient plans to meet elderly women's sexual demands.

The design of the current protocol has been developed according to the cultural and social context of Iran and provides sufficient information about the sexual health demands of elderly women. Therefore, it appears suitable for countries with characteristics similar to Iran, such as Asian countries, particularly those in the Middle East. Additionally, since a comprehensive review (based on the sexual health of elderly women globally) was conducted, and its results will be integrated with the findings of qualitative studies to develop the questionnaire, this questionnaire can also be useful for elderly women in Western countries. These countries can apply the current study's questionnaire with only one psychometric adjustment.

## Strength of the study

Achieving comprehensive data through qualitative study and literature review is the strength of the study.

## Limitations and recommendations of the study

This study will be conducted for the first time in Iran. Due to cultural issues in the country, interviews with these women's husbands will not be conducted. Future studies using elderly women's and men's experiences, either as couples or separately, are recommended. Interviews with them can better clarify their needs and demands, allowing for appropriate interventions

and programs tailored to elderly couples. Research questions could be: As a couple, what demands do you feel are necessary to improve your sexual health in old age? What strategies do you suggest for maintaining and improving your sexual health in old age?

## Conclusions

Sexual health is one of all society members' fundamental rights. The deficiency in this right leads to decreased quality of life and physical, mental, and social injuries resulting from family problems. It seems that identifying elderly women's culture- and society-based sexual health demands can provide the necessary information for healthcare policymakers and planners. This way, suitable intervention programs for sexual health can be designed and implemented according to elderly women's actual demands in order to improve their sexual health.

## Acknowledgments

This paper was extracted from a reproductive health PhD thesis. The researchers would like to express their gratitude to the Student Research Management and the University Ethics Committee.

## Author contributions

**Conceptualization:** Marjan Beigi.

**Data curation:** Mitra Savabi-Esfahani.

**Formal analysis:** Farzaneh Saadatmand, Zahra Heidari.

**Methodology:** Mitra Savabi-Esfahani.

**Software:** Zahra Heidari.

**Supervision:** Marjan Beigi.

**Writing – original draft:** Marjan Beigi.

**Writing – review & editing:** Farzaneh Saadatmand.

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
