## [Decision Letter · Decision Letter 0]

22 Oct 2024

PONE-D-24-36737Development of a conceptual framework of sexual health for married elderly women: An exploratory mixed method studyPLOS ONE

Dear Dr. Beigi,

Thank you for submitting your manuscript to PLOS ONE. After careful consideration, we feel that it has merit but does not fully meet PLOS ONE’s publication criteria as it currently stands. Therefore, we invite you to submit a revised version of the manuscript that addresses the points raised during the review process.

We look forward to receiving your revised manuscript.

Kind regards,

Mojtaba Kordrostami, Ph.D.

Academic Editor

PLOS ONE

Journal Requirements:

Reviewers' comments:

Reviewer's Responses to Questions

**Comments to the Author**

1. Does the manuscript provide a valid rationale for the proposed study, with clearly identified and justified research questions?

Reviewer #1: Partly

Reviewer #2: Yes

2. Is the protocol technically sound and planned in a manner that will lead to a meaningful outcome and allow testing the stated hypotheses?

Reviewer #1: Partly

Reviewer #2: Yes

3. Is the methodology feasible and described in sufficient detail to allow the work to be replicable?

Reviewer #1: No

Reviewer #2: Yes

4. Have the authors described where all data underlying the findings will be made available when the study is complete?

Reviewer #1: No

Reviewer #2: Yes

5. Is the manuscript presented in an intelligible fashion and written in standard English?

Reviewer #1: No

Reviewer #2: Yes

6. Review Comments to the Author

You may also provide optional suggestions and comments to authors that they might find helpful in planning their study.

Reviewer #1: • The language can be refined for clarity. Some sentences are lengthy and may benefit from more direct phrasing. Simplify the text to ensure that the key points are easily understood.

• There are grammatical errors, such as "elimination of agism attitudes" which should be corrected to "elimination of ageist attitudes."

• The subcategories in each group could be elaborated further to provide more detailed explanations or examples. For instance, what specific strategies or practices might be involved in "eliminating sexual superiority for men" or "enabling enjoyment of sexual pleasure in old age"?

• Can you elaborate on the criteria used for selecting the studies included in the review?

• How do you account for demographic variations (e.g., socioeconomic status, education level) in the final questionnaire?

• What specific methods employed to ensure the reliability and validity of the qualitative interviews?

• What ethical guidelines followed to ensure the protection of participants' privacy and confidentiality during qualitative interviews?

• Describe how the 83 studies were identified, including databases searched and search terms used. Specify inclusion and exclusion criteria.

• Detail the process of extracting data from the selected studies and how this data was categorized.

• Provide specifics on the interview process, including participant recruitment, interview structure, and analysis methods.

• Explain the procedures for developing and validating the questionnaire, including pilot testing and statistical analyses to assess reliability and validity.

• The discussion section should delve deeper into the implications of the findings. Explore how the study's results compare to existing knowledge, how they advance understanding of elderly sexual health, and how they can be practically applied.

• Consider discussing barriers to implementing the proposed changes in regions with strong cultural resistance, providing insight into how these challenges could be addressed.

• While the study highlights cultural issues specific to Iran and similar contexts, it could benefit from a more balanced discussion that also includes perspectives from different cultural settings. A comparison could enhance the generalizability of the findings.

• Strengthen the argument that the identified demands are deeply rooted in socio-cultural norms by integrating evidence from other studies or cultural analyses. This approach will provide a more comprehensive understanding of how these issues are shaped and perpetuated.

• Emphasize how addressing sexual health demands of elderly women can lead to broader societal benefits beyond individual well-being, such as strengthening family dynamics and improving community health.

• The limitations mention that interviews with elderly couples could be beneficial, but this point could be expanded. What specific research questions should be explored in these future studies? How could they potentially impact interventions or policies?

Reviewer #2: Dear Authors, I liked the hard work that you have put into this work and that you will be doing in the future, however I would like to be more convinced about the importance of your study, especially when you mention policymakers. I think the reader has to really appreciate this much effort that you are putting into an important subject by the rationales that you are giving as important of your study. For me as a reader, I think you have not added strong rationales for your hard work.

7. PLOS authors have the option to publish the peer review history of their article (what does this mean? ). If published, this will include your full peer review and any attached files.

**Do you want your identity to be public for this peer review?** For information about this choice, including consent withdrawal, please see our Privacy Policy .

Reviewer #1: No

Reviewer #2: No

---

## [Author Response · Author response to Decision Letter 1]

16 Nov 2024

I sincerely thank you for the opportunity to edit, revise and present the article again. And many thanks to the reviewers for their valuable comments that were very helpful in improving the writing of the manuscript.

We have carefully reviewed the comments and have revised the manuscript accordingly. Our answers are given point by point in files of response to Reviewers and manuscript. Changes to the manuscript our shown in highlights (yellow). And the changes in the English version are marked in yellow. We hope revised version is now suitable for publication and looking forward to hearing from you in due course.

Sincerely

Marjan Beigi

---

## [Decision Letter · Decision Letter 1]

11 Dec 2024

PONE-D-24-36737R1Development of a conceptual framework of sexual health for married elderly women: An exploratory mixed method studyPLOS ONE

Dear Dr. Beigi,

Thank you for submitting your manuscript to PLOS ONE. After careful consideration, we feel that it has merit but does not fully meet PLOS ONE’s publication criteria as it currently stands. Therefore, we invite you to submit a revised version of the manuscript that addresses the points raised during the review process.

We look forward to receiving your revised manuscript.

Kind regards,

Mojtaba Kordrostami, Ph.D.

Academic Editor

PLOS ONE

Journal Requirements:

Reviewers' comments:

Reviewer's Responses to Questions

**Comments to the Author**

1. Does the manuscript provide a valid rationale for the proposed study, with clearly identified and justified research questions?

Reviewer #1: Partly

Reviewer #2: No

2. Is the protocol technically sound and planned in a manner that will lead to a meaningful outcome and allow testing the stated hypotheses?

Reviewer #1: Partly

Reviewer #2: Yes

3. Is the methodology feasible and described in sufficient detail to allow the work to be replicable?

Reviewer #1: Yes

Reviewer #2: Yes

4. Have the authors described where all data underlying the findings will be made available when the study is complete?

Reviewer #1: Yes

Reviewer #2: Yes

5. Is the manuscript presented in an intelligible fashion and written in standard English?

Reviewer #1: Yes

Reviewer #2: Yes

6. Review Comments to the Author

You may also provide optional suggestions and comments to authors that they might find helpful in planning their study.

Reviewer #1: 1. While the use of STROBE and COREQ checklists is commendable, specifying how these were adapted or applied in detail would strengthen reproducibility and transparency.

2. Non-random sampling for quantitative phases may introduce selection bias. Discussing how this limitation will be mitigated (e.g., stratification or weighting during analysis) would add value.

3. While Graneheim and Lundman’s approach to content analysis is appropriate, elaborating on strategies to ensure consistency among coders (e.g., inter-coder reliability measures) would improve methodological rigor.

4. The process for engaging policymakers and ensuring their input into tool development could be clarified. For example, incorporating a Delphi method for expert consensus might add depth to the final recommendations.

5. As the study focuses on elderly women within specific cultural contexts (Iranian nationality), discussing how findings might be applicable or adapted to other settings would enhance the study's global relevance.

6. The use of Moghasemi's questionnaire for concurrent validity is innovative, but discussing potential overlaps and ensuring construct distinction between the tools is essential to avoid redundancy.

7. Including feedback loops after presenting demands to policymakers can ensure that the recommendations are actionable and contextually relevant.

8. Plans for the longitudinal assessment of the tool’s impact on sexual health services and outcomes should be considered to validate its practical utility.

Reviewer #2: I appreciate your efforts on this subject, editing the manuscript and addressing the comments. This study will bring insigt for professions in this field.

7. PLOS authors have the option to publish the peer review history of their article (what does this mean? ). If published, this will include your full peer review and any attached files.

**Do you want your identity to be public for this peer review?** For information about this choice, including consent withdrawal, please see our Privacy Policy .

Reviewer #1: No

Reviewer #2: No

---

## [Author Response · Author response to Decision Letter 2]

23 Dec 2024

I sincerely thank you for the opportunity to edit, revise and present the article again. And many thanks to the reviewers for their valuable comments that were very helpful in improving the writing of the manuscript. We have carefully reviewed the comments and have revised the manuscript accordingly

---

## [Editor Report · Decision Letter 2]

10 Jan 2025

Development of a conceptual framework of sexual health for married elderly women: An exploratory mixed method study

PONE-D-24-36737R2

Dear Dr. Beigi,

We’re pleased to inform you that your manuscript has been judged scientifically suitable for publication and will be formally accepted for publication once it meets all outstanding technical requirements.

Kind regards,

Mojtaba Kordrostami, Ph.D.

Academic Editor

PLOS ONE
---

## [Editor Report · Acceptance letter]

PONE-D-24-36737R2

PLOS ONE

Dear Dr. Beigi,

I'm pleased to inform you that your manuscript has been deemed suitable for publication in PLOS ONE. Congratulations! Your manuscript is now being handed over to our production team.

Kind regards,

on behalf of

Dr. Mojtaba Kordrostami

Academic Editor

PLOS ONE
